# Increased and sex-selective avian predation of desert locusts *Schistocerca gregaria* treated with *Metarhizium acridum*

Wim C. Mullié[1]☺¤*, Robert A. Cheke[2]☺, Stephen Young[2], Abdou Baoua Ibrahim[3], Albertinka J. Murk[4]

**1** Emergency Centre for Locust Operations, Food and Agriculture Organization of the United Nations, Via delle Terme di Caracalla, Rome, Italy, **2** Natural Resources Institute, University of Greenwich at Medway, Central Avenue, Chatham Maritime, Kent, United Kingdom, **3** Direction Générale de la Protection des Végétaux (DGPV), Niamey, Niger, **4** Marine Animal Ecology group, Wageningen University and Research, Wageningen, The Netherlands

☺ These authors contributed equally to this work.
¤ Current address: Dakar-Fann, Senegal
* wim_sen@yahoo.fr

**Data Availability Statement:** All relevant data are within the manuscript and its Supporting information files.

**Funding:** The field research of WCM, RAC and ABI was executed under project OSRO/RAF/432/GER

## Abstract

The entomopathogenic fungus *Metarhizium acridum* in oil-based formulations (Green Muscle® (GM)) is a biopesticide for locust control lacking side-effects on biodiversity, unlike chemical insecticides. Under controlled conditions, GM-treated locusts and grasshoppers attract predators, a complementary advantage in locust control. We assessed avian predation on a population of desert locusts in northern Niger aerially sprayed operationally with GM with 107 g viable conidia ha$^{-1}$. Populations of adult locusts and birds and vegetation greenness were assessed simultaneously along two transects from 12 days before until 23 days after treatment. Common kestrels *Falco tinnunculus* and lanners *F. biarmicus* were the predominant avian predators. Regurgitated pellets and prey remains were collected daily beneath "plucking posts" of kestrels. Locusts started dying five days post-spray and GM had its maximum effect one-two weeks after the spray, with 80% efficacy at day 21. After spraying, bird numbers increased significantly ($P<0.05$) concurrent with decreasing desert locust densities. Locust numbers decreased significantly ($P<0.001$) with both time since spraying and decreasing greenness. Before spraying, kestrel food remains under plucking posts accounted for 34.3 ±13.4 prey items day$^{-1}$, of which 31.0 ±11.9 were adult desert locusts (90.3%), reducing post-spray to 21.1 ±7.3 prey items day$^{-1}$, of which 19.5 ±6.7 were adult desert locusts (92.5%), attributable to decreased use of the plucking-posts by the kestrels rather than an effect of the spray. After spraying, kestrels took significantly ($P<0.05$) more larger female (75–80%) than smaller male (20–25%) locusts. Avian predation probably enhanced the impact of the GM on the desert locust population, especially by removing large adult females. No direct or indirect adverse side-effects were observed on non-target organisms including locust predators such as ants and birds. These substantial ecological advantages should also be considered when choosing between conventional chemical and biopesticide-based locust control.

"Assistance to the current desert locust upsurge in the Sahelian countries - Alternative locust control methods", funded by the government of Germany and implemented by the Food and Agriculture Oranization of the United Nations (FAO). The funders had no role in study design, data collection and analysis, decision to publish, or preparation of the manuscript.

**Competing interests:** The authors have declared that no competing interests exist.

## Introduction

Most locusts, including the desert locust *Schistocerca gregaria*, are usually controlled by synthetic pesticides such as the organophosphates fenitrothion, chlorpyrifos and malathion, e.g. during the 2003–2005 desert locust upsurge [1] and during the present desert locust outbreak in Eastern Africa, the Arabian peninsula and SW Asia with over 2 million hectares being sprayed between 1 January 2019 and 31 March 2020 [2]. These pesticides, if applied correctly kill sprayed locusts within hours, however, they also kill or debilitate natural enemies of locusts, such as birds [3] and insects including Coleoptera, Hymenoptera and Diptera [4, 5]. This reduces the efficiency of the overall control by diminishing the impact of the natural predators and parasitoids, and by inducing secondary pests [6]. Also, other beneficial insects are killed, such as pollinators, thus increasing detrimental long-term ecological effects of locust control [7]. The need for alternative approaches that do not kill or debilitate the natural allies was already being advocated decades ago for sustainable locust control [8] and remains a pressing need during the current outbreak in eastern Africa, the Middle East and Pakistan [9].

An alternative to chemical pesticides is the application of biological agents such as entomopathogenic fungi. Formulations of the aerial conidia of isolates of the deuteromycete fungus *Metarhizium acridum* in oil-based suspensions were found to be effective, even in dry environments [10–12]. Following developments by the LUBILOSA program (CABI Bioscience, Ascot, UK) [13], formulations of *M. acridum* were registered as 'Green Muscle$^{®}$' (GM) for use against the brown locust *Locustana pardalina* in South Africa in 1998 and against desert locusts and grasshoppers in nine Sahelian countries in 2001. The genus *Metarhizium* can kill a wide range of insects, but GM is based on a specific isolate which targets locusts and grasshoppers [13]. This specificity is an important feature because the product has little or no adverse environmental impact, benefiting not only humans but also other animals, including the natural enemies of the pests. The Food and Agriculture Organization of the United Nations (FAO), based on the latest recommendation of its Pesticide Referee Group [14] considers biopesticides based on *M. acridum* to be the most appropriate option for locust control, but many African countries still lag behind in the registration process. A second strain of *M.acridum* (EVCH 077), marketed as NOVACRID, was registered in November 2019 by the Comité Sahélien des Pesticides for use in the Sahel.

During trials of GM against hoppers of desert locusts in Mauritania and Algeria, very few dead or dying hoppers were found after the treatment. This was attributed to predation by birds attracted to high densities of sick and sluggish prey [15, 16]. The same phenomenon was observed after a trial with GM against adult red locusts *Nomadacris septemfasciata* in Tanzania [17], suggesting that there might be an interaction between GM and enhanced avian predation on surviving or impaired locusts. Indications of increased bird densities after application of biopesticides also came from trials with the microsporidium *Nosema locustae* [18]. Predation by Hoopoe *Upupa epops* on pupae of the pine processionary caterpillar *Thaumetopoea pityocampa* was enhanced when the prey was infected with *Beauveria bassiana* [19]. Increased densities after application of fungal pesticides have also been reported for invertebrate predators that are apparently unaffected by the fungi, thus enhancing pest mortality [20]. The seemingly enhanced contributions by wild birds to pest reductions upon application of GM contrast with the excess avian mortalities caused by indiscriminate ecotoxicological effects associated with the use of synthetic pesticides [21, 22].

At least 537 species of birds from 61 different families are known to attack acrids in Africa, of which 146 species are known to feed on hoppers and adults of the desert locust, especially as predators of locust swarms [23]. Field data on bird predation on hopper bands and swarms show that birds can regulate locust populations at low and medium densities [23].

Also, experimental studies with grasshoppers using bird exclosures showed the potential of birds to regulate orthopteran populations. It was found that grasshopper densities outside exclosures were 33% lower in a year with average rainfall, and similar in a dry year with low densities [24], whereas a 27.4% reduction in grasshoppers in plots subjected to 40 days of bird predation was reported [25]. In another study, an average annual bird-induced reduction of 25% in a three year experiment was reported [26]. At the end of a 4-year bird exclusion field experiment, adult grasshopper density was 2.2 times higher, and nymph density 3 times higher in exclosures [27]. In the only bird exclosure study done in Africa [28], the authors found that grasshopper densities in Senegal outside exclosures at the end of the rainy season were 30% lower than inside, 37 days after the start of the experiment.

This study quantifies the effects of an operational application of GM against an isolated population of desert locusts on avian locust predation. Our hypotheses were:

1. avian predation complements the impact of the fungal insecticide to control the locusts, and

2. birds prefer the larger females of the desert locust over the males given that, in general, birds preferentially take larger species of grasshoppers or locusts or, within a species, the larger sex [29–31].

## Materials and methods

### Study area

Potential breeding sites for the desert locust were selected during a helicopter survey between Agadez and Arlit in Northern Niger in September 2005. At Aghéliough (18˚ 46'N, 7˚ 31'E), approximately 15 km NNE of Arlit and West of the Aïr mountains, an extensive stand of 530 ha of *c*. 80% green *Schouwia thebaica* was found supporting an isolated population of hoppers and immature adults (>3000 ind. ha$^{-1}$ on 15 September 2005) of transiens phase desert locusts. Freshly laid egg-pods were found. At the site dozens of black-crowned finch larks *Eremopterix nigriceps* and golden sparrows *Passer luteus* were breeding and adults were seen taking hoppers to feed their young.

On 23 October, two weeks before treatment, about 45% of the vegetation was still green. Two perpendicular transects, respectively 3.55 and 3.25 km long, across the area were chosen (Fig 1, resp. A and B) to survey the locust and bird populations.

As both birds and locust numbers were expected to be influenced by vegetation density and greenness, the percentage of the surface covered by green vegetation in a 50 m wide stretch on either side of the transects was classified on a 5-digit scale: 0–0% green (vegetation had died and turned brown), 1–1–25% green, 2–26–50% green, 3–51–75% green and 4–76–100% green.

### GM treatment and locust surveys

The study area was aerially treated with GM (Biological Control Products SA (Pty) Ltd) on 5 November 2005 (08:00–11:00 hrs) with a Cessna 188 fixed wing aircraft, fitted with four Micronair$^{®}$ type AU5000 atomizers (Micron Sprayers Ltd) and the spray tracks recorded by a GPSmap 60CS (Garmin Ltd). The operational application dose of GM was 107 g viable conidia ha$^{-1}$. Spray deposit distribution and intensity (N droplets cm$^{-2}$) were assessed with oil sensitive papers at 5 m intervals, placed 1.5 m high along three lines (C1-C3 on Fig 1), each 200 m long and 800 m apart, perpendicular to diagonal A (SW-NE) of the plot. A fourth line with spray

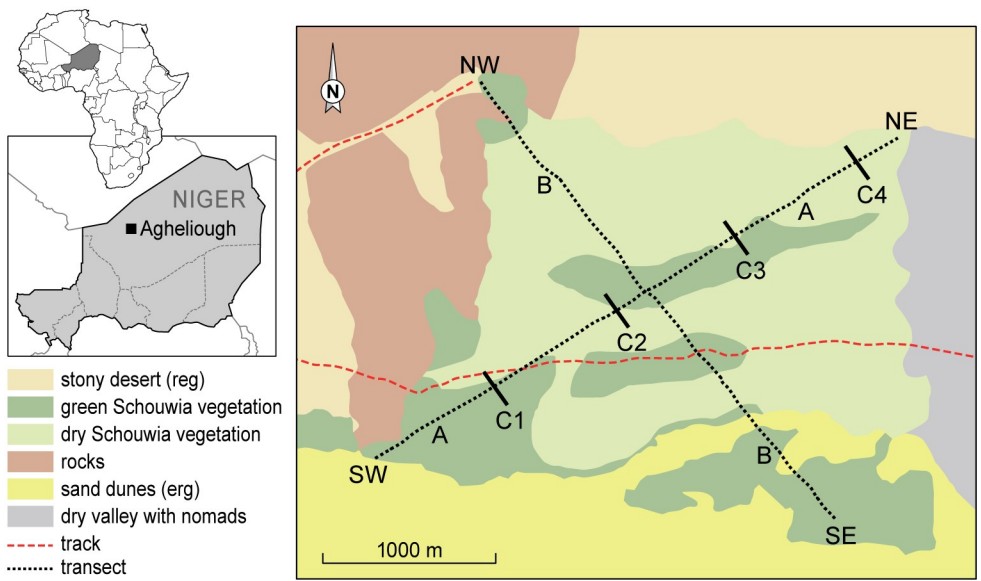

**Fig 1. Map of study site and location of transects A (SW-NE) and B (SE-NW).** The greenness of the *Schouwia* vegetation is the situation by mid-November. The lines C1-C4 indicate the position of the spray papers.

papers (C4) was just outside the sprayed area which was retained for the study. Densities of immature winged desert locusts were estimated every other day by counting all seen within 1 m on either side along each 100 m stretch of the transects, using a tally counter, while walking at a standard pace of c. 0.6 m s$^{-1}$ (2.3 km h$^{-1}$). Pre-treatment counts took place from 24 October until 3 November, and post-treatment from 6 until 28 November. In parallel with our study, the efficacy of the treatment on locusts and the infectivity of *M. acridum* with locusts captured in the field and transferred to cages in the lab was assessed [32]. Relevant outcomes of that study will be referred to in the results section of this paper.

Light traps were set up on 19, 20 and 21 November and operated from 19:00 until 03:00 hrs at 18˚45'33.8"N, 007˚33'50.9"E, 2.4 km E of the SE starting point of transect B (Fig 1). This was to determine whether any locusts were flying away from the treated area and thus contributing to a reduction in numbers on the study site. While driving to and from the light trap, activity of nocturnal locust predators was monitored. This was the only area outside the study site which still contained green vegetation but without desert locusts.

## Locust morphometrics

Before(27 October) and after (10 November) spraying, 164 adult locusts were randomly collected in the vegetation in the early morning while they were still inactive, sexed, weighed and body sizes measured when still fresh.

## Bird censuses and behavioral observations

The same transects that were used for the locusts were surveyed for birds by recording all birds seen within 50 m on either side of the transect. Surveys were conducted according to the same schedule as used for the locusts. Binoculars (10 x 40), and a telescope (10-30x zoom) were used whenever needed. Special attention was given to the behavior of falcons (kestrels *Falco tinnunculus* and lanners *F. biarmicus abyssinicus* and *F. b. erlangeri*), the most obvious locust

predators present. The behavior of other avian locust predators, both diurnal and nocturnal, was only assessed qualitatively.

Between 09:00 and 10:00 hrs on each day from 27 October until 29 November, all pellets and remains of locusts found were collected from beneath two *Acacia raddiana* trees near the crossing of the transects used as "plucking posts" by falcons to consume their locust prey (designated KTREE1 and KTREE2). Before starting the daily collection under KTREE1 and KTREE2, all pellets and plucking remains were removed. Locust species in both the remains and the pellets were identified. Each daily sample was considered representative of what the birds had been consuming there during the previous 24 hour period.

Dry elytra, femurs and tibia recovered were stored and later measured to determine the sex of the locusts taken by the falcons. The drying of body parts slightly reduced their sizes, complicating direct comparison with measurements taken on live locusts. As no fresh desert locusts were available to determine the relative size reductions, 15 females and 18 males of the slightly larger grasshopper species *Ornithacris cavroisi*, later captured at Khelcom, central Senegal, were used to quantify reduction of tibia length after 48 h oven drying at 70˚C. Tibia were chosen because these were sufficiently common in the prey remains to allow for appropriate calculations.

### Statistical procedures and data analysis

All statistics were analyzed with R [33]. Acridivorous bird numbers and corresponding desert locust densities per 100 m transect were analyzed with a Linear Model (multiple regression) taking greenness of the vegetation and time after treatment as covariates. Data from transect sections with zero locusts recorded were excluded from the analysis.

To test the null hypothesis that the larger desert locust females would be preferentially taken by the falcons, analysis of the length—frequency data of tibia and elytra (as a proxy for body size) of desert locusts was performed with the likelihood ratio test for bimodality in univariate two-component normal mixtures from [34]. The package allows for equal as well as distinct variances, and accounts for finite sample corrections based on penalizing variance estimates under the alternative.

## Results

### Quality of treatment and direct effect of GM on locusts

GM spray deposits on oil sensitive papers were evenly distributed (range 5–60, mean. *c.* 15 droplets cm$^{-2}$ along the first three lines, C1-C3 (Fig 1)). As the fourth line (C4) was just outside the sprayed area, it was only slightly exposed, showing the precision of the treatment. As shown in the parallel study [32], mortality of locusts started five days after treatment, which was confirmed with caged locusts and by the characteristic red color appearing in freshly dead infected individuals. The maximum effect (mortality rate) was between one and two weeks post-spray. Overall efficacy on day 21 post-spray was 80% and > 90% at the end of the study [32].

### Locust morphometrics and densities

Males (N = 30) were smaller than females (N = 24) with their mean fresh weight (WW) being 58% of that of females. Dry weight (DW) of males was 55% and DW of females was 56% of WW after air drying. Mean WW for combined sexes was 2.25 ±0.92 g (Table 1, which includes morphometric data of all 164 captured individuals).

**Table 1. Morphometrics (mm) and body mass (WW, g) for desert locusts collected at the study site on 27 October and 10 November 2005.**

| Sex | | Body Mass | Femur length | Elytron length |
|---|---|---|---|---|
| | | (WW, g) | F (mm) | E (mm) |
| **Males** | mean (±s.d.) | 1.70 (0.47) | 25.40 (1.07) | 53.11 (2.05) |
| | N | 30 | 85 | 85 |
| **Females** | mean (±s.d.) | 2.93 (0.88) | 29.51 (1.23) | 62.24 (2.20) |
| | N | 24 | 79 | 79 |

The parallel assessment of efficacy [32] showed that throughout the study >99% of the population consisted of adults, the remainder being third to sixth instar nymphs, predominantly fifth. Therefore, the locust density calculations are further based on adult locusts.

The number of adult desert locusts (means and maxima $ha^{-1}$) before and after the treatment are presented in Table 2. Maximum densities ranged from 2250 to 11700 individuals $ha^{-1}$ before spraying. Starting from about five days post spray, locust numbers declined rapidly and became as low as 13 ind. $ha^{-1}$, with a maximum of 200 ind. $ha^{-1}$ at the end of the study.

## Observations of affected locusts and of avian locust predation

Searches for dead locusts revealed very few, and it was obvious that dead insects rapidly disappeared. Presumably nocturnal predation and scavenging activity was responsible for removal of dead or dying insects whereas ants, in particular, took care of the remains. Sluggish dying

**Table 2. Number of adult desert locusts counted on two transects and estimates of mean (± s.d.) and maximum number of desert locusts per ha before and after treatment.**

| Date | Transect B (SE-NW) | | | | Transect A (NE-SW) | | | |
|---|---|---|---|---|---|---|---|---|
| | N counted | mean N/ha | s.d. | max N/ha | N counted | mean N/ha | s.d. | max N/ha |
| *Before treatment* | | | | | | | | |
| 24-Oct-05 | 803 | 1147 | 1530 | 5500 | 858 | 1226 | 1987 | 7750 |
| 26-Oct-05 | 756 | 1080 | 1422 | 6750 | 749 | 1070 | 1125 | 4000 |
| 28-Oct-05 | 307 | 439 | 608 | 2250 | 709 | 1013 | 1514 | 6000 |
| 30-Oct-05 | 442 | 631 | 1000 | 4200 | 1726 | 2466 | 3289 | 11700 |
| 1-Nov-05 | 948 | 1354 | 2057 | 8850 | 1123 | 1604 | 1964 | 6350 |
| 3-Nov-05 | 625 | 893 | 1577 | 7800 | 748 | 1069 | 1377 | 4450 |
| *After treatment* | | | | | | | | |
| 6-Nov-05 | 485 | 693 | 1016 | 3200 | 622 | 889 | 1277 | 5550 |
| 8-Nov-05 | 456 | 651 | 1176 | 5300 | 660 | 943 | 1175 | 5650 |
| 10-Nov-05 | 528 | 754 | 1025 | 3450 | 763 | 1090 | 1165 | 5050 |
| 12-Nov-05 | 257 | 367 | 542 | 1700 | 510 | 729 | 867 | 3550 |
| 14-Nov-05 | 124 | 177 | 239 | 750 | 456 | 651 | 739 | 3250 |
| 16-Nov-05 | 115 | 164 | 240 | 900 | 296 | 423 | 528 | 2400 |
| 18-Nov-05 | 34 | 49 | 119 | 500 | 72 | 103 | 156 | 600 |
| 20-Nov-05 | 30 | 43 | 100 | 450 | 68 | 97 | 150 | 650 |
| 22-Nov-05 | 21 | 30 | 82 | 450 | 56 | 80 | 158 | 700 |
| 24-Nov-05 | 9 | 13 | 33 | 150 | 41 | 59 | 96 | 350 |
| 26-Nov-05 | 13 | 19 | 43 | 200 | 26 | 37 | 56 | 200 |
| 28-Nov-05 | 9 | 13 | 28 | 100 | 20 | 29 | 59 | 200 |

animals, which moved to the upper vegetation layer to bask, to induce behavioral fever [35] as a reaction to the *M. acridum* infection, were rarely encountered.

The two falcon species were regularly observed preying on adult locusts. Examination of photographs showed that, although never more than six different lanners were ever observed on the same day, some individuals disappeared and new ones appeared. *F. b. abyssinicus* was the more abundant subspecies before treatment, whereas *F. b. erlangeri* became more numerous afterwards. Lanners dived from >30m to snatch adult locusts flying about 1m above the *Schouwia* vegetation and exploited locusts that were disturbed into flight by people, camels and goats. Kestrels hovered to detect their prey.

Other bird predators of locusts included crested larks *Galerida cristata*, wheatears *Oenanthe* spp. and southern great grey shrikes *Lanius meridionalis leucopygos*. The latter impaled locusts on thorns of *Acacia raddiana*.

## Bird censuses

The number and species of birds recorded along the two transects before and after spraying are listed in S1 Appendix. In total, 28 species of birds were recorded. Many of these were irrelevant to the locust study as they were aerial feeders (e.g. swifts and swallows) while some acridivorous species such as nubian bustards *Neotis nuba* were never recorded during transect counts. Details of the 16 relevant acridivores are given in Table 3.

The numbers of acridivorous birds recorded along the transects varied from 0.2–20.3 individuals ha$^{-1}$ and interestingly the highest numbers were recorded about 10 days after the treatment. As can be seen in Table 2, this coincided with the peak of the GM effectiveness in terms of the greatest reduction of numbers of locusts counted (between 10 and 24 November). Multiple regression analysis showed significant increases in bird numbers with decreasing locust densities ($P<0.05$), decreasing locust densities with decreasing greenness ($P<0.001$) and decreases in locust densities with increasing time after the spray ($P<0.01$). The model was: Log (locusts) = 2.26 ($\pm$ 0.091) $-$ 0.125 ($\pm$ 0.061) log (total birds) $-$ 0.084 ($\pm$ 0.006) days after the

**Table 3. Total number, frequency and mean ± s.d. of the most common acridivorous birds observed along the transects.**

| Species | scientific name | before treatment (N = 5 counts) | | | | after treatment (N = 12 counts) | | | |
|---|---|---|---|---|---|---|---|---|---|
| | | total | freq. | mean N ha$^{-1}$ | s.d. | total | freq. | mean N ha$^{-1}$ | s.d. |
| Common Kestrel | *Falco tinnunculus* | 30 | 1.00 | 6 | 1.4 | 40 | 0.83 | 3.3 | 2.8 |
| Lanner | *Falco biarmicus* | 3 | 0.60 | 0.6 | 0.5 | 33 | 0.67 | 2.8 | 2.8 |
| Common Quail | *Coturnix coturnix* | not observed | | | | 2 | 0.08 | 0.2 | 0.6 |
| African Hoopoe | *Upupa epops senegalensis* | 5 | 0.60 | 1 | 1.2 | not observed | | | |
| Crested Lark | *Galerida cristata* | 73 | 1.00 | 14.6 | 4.4 | 244 | 0.92 | 20.3 | 16.0 |
| Desert Lark | *Ammomanes deserti* | 1 | 0.20 | 0.2 | 0.4 | not observed | | | |
| Greater Short-toed Lark | *Calendrella brachydactyla* | 37 | 0.80 | 7.4 | 5.5 | 21 | 0.25 | 1.8 | 4.9 |
| Greater Hoopoe Lark | *Alaemon alaudipes* | 2 | 0.40 | 0.4 | 0.5 | not observed | | | |
| Black-eared Wheatear | *Oenanthe hispanica* | 1 | 0.20 | 0.2 | 0.4 | not observed | | | |
| Desert Wheatear | *Oenanthe deserti* | 43 | 1.00 | 8.6 | 4.8 | 109 | 0.92 | 9.1 | 8.7 |
| Isabelline Wheatear | *Oenanthe isabellina* | not observed | | | | 71 | 1.00 | 5.9 | 3.1 |
| Wheatear sp. | *Oenanthe sp.* | not observed | | | | 76 | 1.00 | 6.3 | 4.6 |
| Cricket Warbler | *Spiloptila clamans* | 12 | 0.80 | 2.4 | 1.9 | 4 | 0.17 | 0.3 | 0.9 |
| Fulvous Babbler | *Turdoides fulva* | 6 | 0.40 | 1.2 | 2.2 | not observed | | | |
| Southern Grey Shrike | *Lanius meridionalis leucopygos* | 15 | 1.00 | 3 | 2.5 | 36 | 0.92 | 3.0 | 1.8 |
| Chestnut-bellied Starling | *Lamprotornis pulcher* | 2 | 0.20 | 0.4 | 0.9 | 4 | 0.25 | 0.3 | 0.7 |
| Sudan Golden Sparrow | *Passer luteus* | 38 | 0.60 | 7.6 | 9.1 | 57 | 0.58 | 4.8 | 6.6 |

spray + 0.2745 (± 0.045) greenness (parameter values ±standard errors) (F = 83.44 (3 & 504 DF), $P \ll 0.00001$). In absolute terms, days after the spray had the greatest effect on locust numbers.

After the spraying, on the days that the light trap was operated, some nocturnal predation was observed. In the light of a vehicle's headlights, a lanner was seen hunting, as were spotted thick-knee *Burhinus capensis*, gerbils *Gerbillus* spp. and pale sand fox *Vulpes pallida*. Another nocturnal predator, a wild cat *Felis silvestris libyca* was seen in broad daylight during the spray.

## Indirect observations of avian predation of locusts

The total and daily average prey numbers taken by the kestrels using the plucking posts KTREE1 and KTREE2 before and after the treatment are given in Table 4 and S2 Appendix. Six larger pellets in the bulk material removed before daily collection started were significantly different from the kestrel pellets (Welch two-sample t-Test: t = 9.6049, df = 7.796 $P = 1.367e^{-0.5}$), but not different from lanner pellets containing bird and desert locust remains from Egypt and Sudan [36] (Welch modified two-sample t-Test: t = 1.9773, df = 37.449, N.S.). No such pellets were found during the daily collection of food remains. Therefore, the pellets and prey remains that we recovered were considered to be exclusively from kestrels.

As pellets and plucking remains potentially concerned the same locusts, pellet information was only used to calculate daily consumption when the total numbers of desert locusts were higher in pellets than those in the plucking remains. Coleopteran, mole cricket and bird remains were only present in pellets. Before the spray, in the Kestrel food remains 31.0±11.9 prey items day$^{-1}$ out of a total of 34.3±13.4 or 90.3% were adult desert locusts. After spraying this was 92.5%. In terms of biomass this was 89.8% before and 87.3% after spraying (Table 4).

The frequency distributions of lengths of tibia and elytra recovered under KTREE1 and KTREE2 are significantly bimodal (Likelihood ratio test tibia $P = 0.03$, elytra $P = 0.012$, Table 5; Fig 2 shows the data for elytra). After separating them into two normally distributed

**Table 4. Kestrel prey remains (including from pellets; see text) by number and their calculated biomass (as total wet wt) collected under KTREE1 and KTREE2 before (N = 9 days) and after (N = 24 days) treatments.**

| | Before treatment (N = 9 days) | | | | | After treatment (N = 24 days) | | | | |
|---|---|---|---|---|---|---|---|---|---|---|
| *Taxa by number* | freq. | Total | Daily avg | Std | % | freq. | Total | Daily avg | Std | % |
| desert locust (ad.) | 1 | 290 | 32.22 | 11.99 | 90.6 | 1 | 468 | 19.50 | 6.71 | 92.5 |
| desert locust (5th instar) | 0.11 | 10 | 1.11 | 3.33 | 3.1 | 0.04 | 1 | 0.04 | 0.20 | 0.2 |
| tree locust | 0.56 | 7 | 0.78 | 0.83 | 2.2 | 0.71 | 28 | 1.17 | 1.09 | 5.5 |
| unidentified grasshopper | 0.22 | 7 | 0.78 | 1.72 | 2.2 | 0.04 | 3 | 0.13 | 0.61 | 0.6 |
| mole cricket | 0.22 | 4 | 0.44 | 0.88 | 1.3 | 0 | 0 | 0.00 | 0.00 | 0.0 |
| coleopteran sp. | 0.11 | 1 | 0.11 | 0.33 | 0.3 | 0.17 | 4 | 0.17 | 0.38 | 0.8 |
| small bird | 0.11 | 1 | 0.11 | 0.33 | 0.3 | 0.08 | 2 | 0.08 | 0.28 | 0.4 |
| **Total** | | **320** | **35.55** | **13.13** | **100.0** | | **506** | **21.08** | **7.26** | **100.0** |
| *Biomass (g wet wt)* | BM (g) | Total | Daily avg | Std | % | BM (g) | Total | Daily avg | Std | % |
| desert locust | 2.25 | 652.5 | 72.50 | 26.97 | 90.2 | 2.25 | 1053 | 43.88 | 15.10 | 87.3 |
| desert locust (5th instar) | 1.1 | 11 | 1.22 | 3.67 | 1.5 | 1.1 | 1.1 | 0.05 | 0.22 | 0.1 |
| tree locust | 3.9 | 27.3 | 3.03 | 3.25 | 3.8 | 3.9 | 109.2 | 4.55 | 4.25 | 9.1 |
| unidentified grasshopper | 0.68 | 4.76 | 0.53 | 1.17 | 0.7 | 0.68 | 2.04 | 0.09 | 0.42 | 0.2 |
| mole cricket | 2 | 8 | 0.89 | 1.76 | 1.1 | 2 | 0 | 0.00 | 0.00 | 0.0 |
| coleopteran sp. | 0.1 | 0.1 | 0.01 | 0.03 | 0.0 | 0.1 | 0.4 | 0.02 | 0.04 | 0.0 |
| small bird | 20 | 20 | 2.22 | 6.67 | 2.8 | 20 | 40 | 1.67 | 5.65 | 3.3 |
| **Total** | | **723.66** | **80.41** | **27.56** | **100.0** | | **1205.74** | **50.24** | **17.38** | **100.0** |

**Table 5. Mean length (mm) ± s.d. of tibiae and elytra of male and female desert locusts captured alive and in prey remains.** For length adjustments because of drying of prey remains, see text. The bimodal distribution of prey remains over the sexes has been separated by applying the likelihood ratio test [34].

| | | captured alive | | | | prey remains | | | |
|---|---|---|---|---|---|---|---|---|---|
| | N | length (mm) | proportion | Adjusted length (mm) | N | length (mm) | proportion | likelihood ratio | *p*-value |
| **TIBIA** | 54 | | | | 287 | | | 3.729 | 0.0267 |
| Males | 30 | 23.1 ± 0.89 | 0.55 | 21.7 ± 0.84 | | 22.8 ± 0.89 | 0.23 | | |
| Females | 24 | 27.3 ± 1.38 | 0.45 | 25.7 ± 1.29 | | 25.9 ± 1.83 | 0.77 | | |
| **ELYTRA** | 164 | | | | 526 | | | 5.03 | 0.0125 |
| Males | 85 | 53.1 ± 2.05 | 0.52 | 50.0 ± 1.90 | | 50.5 ± 1.90 | 0.26 | | |
| Females | 79 | 62.2 ± 2.20 | 0.48 | 58.6 ± 2.10 | | 58.0 ± 3.39 | 0.74 | | |

components, the resulting distributions overlap with those of the sample of males and females which were collected during the study. Therefore, we attribute the two distributions to males and females. The bimodality test shows that kestrels took significantly more female than male locusts. Based on the larger sample of elytra lengths the proportion of females taken as prey was 0.74 while this proportion in the sample of live captured adult desert locusts was 0.48 (Table 5).

## Discussion

The results showed that Green Muscle® containing the entomopathogenic fungus *Metarhizium acridum* had strongly reduced locust numbers starting about five days after spraying (Table 2), without any negative impact on acridivorous bird numbers (Table 3) which continued feeding on locusts, including impaired individuals.

Our finding that acridivorous birds were not affected by GM, bird numbers being even higher after treatment than they were before spraying, is in accordance with the finding that chicks of ring-necked pheasant *Phasianus colchicus* fed with *M. acridum* infected grasshoppers

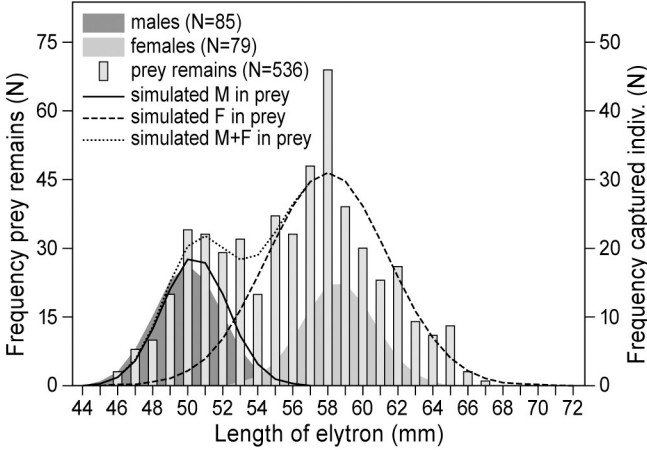

**Fig 2. Frequency distributions of elytra of captured male (N = 85) and female (N = 79) desert locusts.** Superimposed are frequency distributions (bars) of elytra (N = 536) recovered beneath kestrel plucking posts, separated into two complementary normal distributions (males solid line and females broken line). The dotted line is the sum of the male and female normal distributions. The distribution of lengths of elytra from captured locusts was significantly bimodal, *P* = 0.012 [34].

were unaffected [37]. An increase of acridivorous birds in plots treated with entomopathogens against grasshoppers has been reported before [18, 38]. This is in sharp contrast with reports from application of synthetic pesticides such as chlorpyrifos and fenitrothion, widely used for desert locust control. Such conventional chemical insecticides kill a large proportion of the target as well as non-target species within a short period (usually <24 hours) [5]. This includes natural locust predators or parasitoids of locusts or their eggs, such as birds, hymenopterans, coleopterans and arachnids. The dead insects remaining rapidly decay and become unattractive as prey or carrion [39]. Only a few insects not killed by the spray are available as prey and birds temporarily leave the treated zones [3]. In plots treated with chlorpyrifos and fenitrothion, approximately 50% of the bird numbers present just before treatment had moved away within 24 hours post spray [3]. This means that newly arriving or surviving locusts hardly encounter natural enemies anymore, potentially leading to a secondary pest outbreak [6]. In addition, synthetic pesticide applications can also damage other ecosystem functions and insecticide residues may pose risks for the avian and even human food chain [40, 41]. Many desert locust treatments are carried out in remote arid areas and side effects on birds and other beneficial organisms are rarely monitored [23] despite the existence of FAO recommendations [42]. This will probably result in underestimates of the impacts of synthetic pesticides applied for locust or grasshopper control [40] in the longer term.

A fundamental difference between the synthetic and selective biopesticides is that the latter do not kill instantly. While the fungal infection develops further, the affected insects become sluggish and eventually attach themselves to the upper layers of vegetation to bask and induce fever as a response to the infection, a type of behavioral thermoregulation [35]. This, however, exposes them to predators [43]. In trials in Algeria and Mauritania avian predators were shown to eliminate GM treated desert locust hopper bands within a few days, whereas untreated nymphs persisted [16, 44]. Birds attacked hoppers even before any external effect of the entomopathogen was seen by the observers.

The reduction of locust numbers after GM treatment in our study took several weeks (Table 2), clearly longer than would be the case with synthetic insecticides, but the number of acridivorous birds even increased. These natural enemies clearly continued to feed on locusts, including those affected by the spray. The fact that GM neither kills nor debilitates natural enemies of locusts or contribute to their temporary immigration implies that these remain in the system and continue consuming locusts, including new arrivals.

Few studies have reported how natural predators continued to prey upon locusts affected by *Metarhizium*. Ants and beetles were observed to take cadavers of red locust tinted red by *M. acridum* infection 12–15 days after a GM application in Katavi National Park in Tanzania [17, 45]. Within three days 40 dead red locusts, that were placed in the Iku plains treated with 50 g ha$^{-1}$ of GM, almost all disappeared. Sick and dying grasshoppers and locust hoppers, rendered sluggish by the *Metarhizium*, were also seen being taken by birds and frogs (R. E. Price, pers. comm., November 2005). Locust remains found in droppings indicated that some mammals also had eaten many locusts [17, 45].

In the present study it was very difficult to recover dead or dying locusts in the dense *Schouwia* vegetation, although ants were observed to drag locust remains to their nests from out of this vegetation. In an earlier study it was also stated that finding cryptically colored locust cadavers amongst the dense grass was almost impossible [17, 45]. Probably most predation remained unseen because it took place at night. A lanner was seen hunting by night, a behavior which has been seen in kestrels [46] and suggested for lanners [47].

Based on our daily observations, we are confident that only two kestrels used the plucking posts before spraying, where remains of at least 20–59 locusts were recovered daily, and it is likely that the kestrels were regularly satisfying their daily energy requirements from locusts

alone (Table 4). After the treatment, up to two lanners frequently used the trees as roosts. During our daily monitoring of the plucking posts, lanners were never seen plucking prey in the trees, but frequently did so elsewhere on the ground or in the air. The presence of these larger falcons may have restricted the use by kestrels of the plucking posts by kestrels, thereby limiting the number of remains that could be recovered. Pellets were not always found on daily searches and the number of desert locusts in the plucking remains fluctuated widely: 5–34 individuals day$^{-1}$ after the spray. Numbers of prey derived from the remains should be considered as minima and the difference between pre- and post-spray items rather as an indication of changes in the use of the trees by kestrels than of changes in food intake.

Based on observations, we estimated that at least 10 kestrels and six lanners were daily present on the plot throughout the study. The number of locusts eaten by these birds initially only represented a small proportion of the locusts present before spraying but became more important when locust numbers decreased after treatment and predation continued. In contrast to the kestrels, the lanners only took locusts on the wing, presumably the insects least affected by the biopesticide.

The remains of locusts beneath the plucking posts reveal that kestrels preferentially caught females (Table 5, Fig 2), given their larger size this means energy maximization of their attacks [29], confirming our initial hypothesis. In experimental studies of grasshoppers, it has been shown that birds specifically select the larger species, or within a species the larger sex [29, 30, 38], unless the behavior of the males exposed them more than the females [48].

Our study revealed that GM can be used effectively in controlling adult desert locusts, with the important added advantage, in contrast with classical synthetic insecticides, that it does not kill or debilitate birds and other locust predators but rather facilitates them. The lethal effect of *M. acridum* is delayed, but its infectivity was found to last for up to two months under field conditions in Senegal at the end of the rainy season [28]. This implies that newly arriving (sub)adult locusts and locust hatchlings emerging in treated fields within that period may also become infected. This strongly contrasts with synthetic insecticides which rapidly decay on vegetation [40] and do not guarantee any longer lasting effect on newly arriving locusts or emerging nymphs, while causing ecotoxicological side-effects and potentially human health risks. The observed combination of significant increases in acridivorous bird numbers without having a negative impact on other locust predators, both diurnal and nocturnal, has the potential to further enhance the long-lasting effect of GM treatments, offering substantial additional advantages for selective locust control operations.

## Supporting information

**S1 Appendix. Birds recorded at Aghéliough (plus records from Arlit only).**
(DOCX)

**S2 Appendix. Prey remains found under plucking posts K1TREE and K2TREE.**
(XLSX)

## Acknowledgments

We are grateful to the Food and Agriculture Organization of the United Nations (FAO) for the opportunity to conduct this study and to the Office for Corporate Communication for clearance of this article for publication. In Niger, we were aided by staff of the FAO Regional Representation, the Direction de Protection des Végétaux including the pilot of the spray aircraft Christian Collomb, the Direction Régionale de l'Environnement d'Agadez and the AGRHYMET Regional Centre. Much assistance and advice was also provided by the late Dr Clive

Elliott, Dr James Everts and Keith Cressman and other staff at the FAO HQ in Rome. We kindly acknowledge the comments of Dr Amulen Deborah Ruth and the preparation of the figures by Mr Dick Visser, University of Groningen, The Netherlands.

The views expressed in this article are those of the author(s) and do not necessarily reflect the views or policies of the Food and Agriculture Organization of the United Nations.

## Author Contributions

**Conceptualization:** Wim C. Mullié, Robert A. Cheke.

**Data curation:** Wim C. Mullié, Robert A. Cheke, Stephen Young.

**Formal analysis:** Wim C. Mullié, Robert A. Cheke, Stephen Young.

**Investigation:** Wim C. Mullié, Robert A. Cheke, Abdou Baoua Ibrahim.

**Methodology:** Wim C. Mullié, Robert A. Cheke.

**Writing – original draft:** Wim C. Mullié, Robert A. Cheke, Albertinka J. Murk.

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
