## [Decision Letter · Decision Letter 0]

23 Oct 2020

PONE-D-20-15116

Increased and sex-selective avian predation of Desert Locusts Schistocerca gregaria treated with Metarhizium acridum (Green Muscle®)

PLOS ONE

Dear Dr. Mullié,

Thank you for submitting your manuscript to PLOS ONE. After careful consideration, we feel that it has merit but does not fully meet PLOS ONE’s publication criteria as it currently stands. Therefore, we invite you to submit a revised version of the manuscript that addresses the points raised during the review process.

We look forward to receiving your revised manuscript.

Kind regards,

Guy Smagghe, PhD

Academic Editor

PLOS ONE

Journal Requirements:

2. During your revisions, please note that a simple title correction is required. For legal reasons, we no longer allow registered trademark names and symbols in article titles. Please amend your title to "Increased and sex-selective avian predation of Desert Locusts Schistocerca gregaria treated with an entomopathogenic fungus". Please ensure this is updated in the manuscript file and the online submission information. Thank you for your understanding.

3.We note that [Figure(s) 1] in your submission contain [map/satellite] images which may be copyrighted. All PLOS content is published under the Creative Commons Attribution License (CC BY 4.0), which means that the manuscript, images, and Supporting Information files will be freely available online, and any third party is permitted to access, download, copy, distribute, and use these materials in any way, even commercially, with proper attribution. For these reasons, we cannot publish previously copyrighted maps or satellite images created using proprietary data, such as Google software (Google Maps, Street View, and Earth). For more information, see our copyright guidelines: http://journals.plos.org/plosone/s/licenses-and-copyright.

1.    You may seek permission from the original copyright holder of Figure(s) [1] to publish the content specifically under the CC BY 4.0 license. 

Reviewers' comments:

Reviewer's Responses to Questions

**Comments to the Author**

1. Is the manuscript technically sound, and do the data support the conclusions?

Reviewer #1: Yes

2. Has the statistical analysis been performed appropriately and rigorously? 

Reviewer #1: Yes

3. Have the authors made all data underlying the findings in their manuscript fully available?

Reviewer #1: Yes

4. Is the manuscript presented in an intelligible fashion and written in standard English?

Reviewer #1: Yes

5. Review Comments to the Author

Reviewer #1: The topic of the study is very significant considering the recent devastating effects of the desert locusts in the African continent.

The study was well structured, experiments setup as per standards. Authors also analyzed their data sufficiently.

Revisions:

The sentence in line 126 to 129 seems too long, maybe they could shorten.

In results section: some of the results were described after the table, it seemed confusing. It would be better if all results are described before table citation.

Besides the above two comments, I find the manuscript sound with significant findings that can contribute towards sustainable control of locust in Africa and globe.

6. PLOS authors have the option to publish the peer review history of their article (what does this mean?). If published, this will include your full peer review and any attached files.

Reviewer #1: **Yes: **Amulen Deborah Ruth

---

## [Author Response · Author response to Decision Letter 0]

6 Nov 2020

We are very grateful to the useful comments which will make the paper better understandable.

A. Editors remarks:

1. PlosOne's style requirements. We have verified the style and we believe it complies with the requirements.

2. Following the previous correspondence with the academic editor, we have removed the commercial name from the title but kept the scientific name of the fungus as this is a general species name and not linked to a specific isolate which has been commercialized. The title is now: "Increased and sex-selective avian predation of Desert Locusts Schistocerca gregaria treated with Metarhizium acridum".

3. The figure has been completely redrawn and the detailed map is no longer based on any existing cartography but it has been prepared by using our own field records and GPS coordinates.

B. Reviewers responses:

We thank the reviewer for her kind words as far as the importance of the paper is in relation to the ongoing devastating Desert Locust outbreak in Eastern Africa. We are grateful to her suggestions to improve the quality of the paper. Our replies are as follows:

1. no modifications necessary

2. no modifications necessary

3. no modifications necessary

4. "The sentence in line 126 to 129 seems too long, maybe they could shorten".

This has been revised as follows: [new lines 124 to128]

" Our hypotheses were: 

(1) avian predation complements the impact of the fungal insecticide to control the locusts, and 

(2) birds prefer the larger females of the desert locust over the males given that, in general, birds preferentially take larger species of grasshoppers or locusts or, within a species, the larger sex [29-31]".

In results section: "some of the results were described after the table, it seemed confusing. It would be better if all results are described before table citation".

 We agree with the comments and all results are now placed before the figure.

C. In addition to the above mentioned modifications we have made a few minor changes :

In the introduction: The phrase "Although GM is no longer......marketed as NOVACRID (http://en.elephant......novacrid/) has been replaced by: 

"A second strain of M.acridum (EVCH 077), marketed as NOVACRID, was registered in November 2019 by the Comité Sahélien des Pesticides for use in the Sahel". [new lines 92 and 93]

The caption of Fig 1 has been replaced by: 

"Figure 1. Map of study site and location of transects A (SW-NE) and B (SE-NW). The greenness of the Schouwia vegetation is the situation by mid-November. The lines C1-C4 indicate the position of the spray papers". 

In the acknowledgements we have added: 

"We kindly acknowledge the comments of Dr Amulen Deborah Ruth on an earlier version of the manuscript and the preparation of the figures by Mr Dick Visser, University of Groningen, The Netherlands".

We trust that we have properly addressed all issues raised either by the editor or the reviewer

---

## [Editor Report · Decision Letter 1]

16 Dec 2020

Increased and sex-selective avian predation of Desert Locusts Schistocerca gregaria treated with Metarhizium acridum

PONE-D-20-15116R1

Dear Dr. Mullié,

We’re pleased to inform you that your manuscript has been judged scientifically suitable for publication and will be formally accepted for publication once it meets all outstanding technical requirements.

Kind regards,

Guy Smagghe, PhD

Academic Editor

PLOS ONE
---

## [Editor Report · Acceptance letter]

21 Dec 2020

PONE-D-20-15116R1 

Increased and sex-selective avian predation of Desert Locusts *Schistocerca gregaria* treated with *Metarhizium acridum*

Dear Dr. Mullié:

I'm pleased to inform you that your manuscript has been deemed suitable for publication in PLOS ONE. Congratulations! Your manuscript is now with our production department. 

Kind regards, 

on behalf of

Prof. Guy Smagghe 

Academic Editor

PLOS ONE